# Classification of 3D Point Clouds Using Color Vegetation Indices for Precision Viticulture and Digitizing Applications

**Francisco-Javier Mesas-Carrascosa** [1,*] [ID], **Ana I. de Castro** [2] [ID], **Jorge Torres-Sánchez** [2] [ID],
**Paula Triviño-Tarradas** [1] [ID], **Francisco M. Jiménez-Brenes** [2] [ID], **Alfonso García-Ferrer** [1] [ID] and
**Francisca López-Granados** [2]

1   Department of Graphic Engineering and Geomatics, University of Cordoba, Campus de Rabanales,
    Crta. IV, km. 396, E-14071 Córdoba, Spain; ig2trtap@uco.es (P.T.-T.); agferrer@uco.es (A.G.-F.)
2   Imaping Group, Department of Crop Protection, Institute for Sustainable Agriculture (IAS), Spanish
    National Research Council (CSIC), E-14004 Córdoba, Spain; anadecastro@ias.csic.es (A.I.d.C.);
    jtorres@ias.csic.es (J.T.-S.); fmjimenez@ias.csic.es (F.M.J.-B.); flgranados@ias.csic.es (F.L.-G.)
*   Correspondence: ig2mecaf@uco.es

**Abstract:** Remote sensing applied in the digital transformation of agriculture and, more particularly, in precision viticulture offers methods to map field spatial variability to support site-specific management strategies; these can be based on crop canopy characteristics such as the row height or vegetation cover fraction, requiring accurate three-dimensional (3D) information. To derive canopy information, a set of dense 3D point clouds was generated using photogrammetric techniques on images acquired by an RGB sensor onboard an unmanned aerial vehicle (UAV) in two testing vineyards on two different dates. In addition to the geometry, each point also stores information from the RGB color model, which was used to discriminate between vegetation and bare soil. To the best of our knowledge, the new methodology herein presented consisting of linking point clouds with their spectral information had not previously been applied to automatically estimate vine height. Therefore, the novelty of this work is based on the application of color vegetation indices in point clouds for the automatic detection and classification of points representing vegetation and the later ability to determine the height of vines using as a reference the heights of the points classified as soil. Results from on-ground measurements of the heights of individual grapevines were compared with the estimated heights from the UAV point cloud, showing high determination coefficients ($R^2 > 0.87$) and low root-mean-square error (0.070 m). This methodology offers new capabilities for the use of RGB sensors onboard UAV platforms as a tool for precision viticulture and digitizing applications.

**Keywords:** UAV imagery; grapevine height; DSM; RGB sensor; structure; vineyard

## 1. Introduction

Precision agriculture involves the collection and use of large amounts of georeferenced data relating to crops and their attributes in production areas at a high spatial resolution [1]. Its purpose is for site-specific management of crop heterogeneity at both time and spatial scales [2] to optimize agricultural inputs. As a second consequence, this high-quality information can be connected with the global objective of creating opportunities for digitizing agriculture by renewing processes and technologies to make the sector more insight-driven and efficient, without forgetting the need for improving yield and final product quality and reducing the environmental impact of agricultural activity. Factors like soil, water availability, pests (e.g., incidences related to weeds, fungi, insects, root-knot nematodes), presence of cover crops between grape rows, topography, and variable climatic

conditions cause different responses in the crop which are ultimately reflected in spatial fluctuations in yield and grape composition [3]. Precision viticulture (PV) and digitizing-related strategies, which fall inside precision agriculture and the wider concept of digital transformation of agriculture, could contribute to solving and managing this spatial heterogeneity in a sustainable way. This is because their main objectives are the monitoring of vineyard variability and the design of site-specific management accordingly to improve production efficiency with reduced inputs (e.g., labor, fuel, water, canopy management, or phytosanitary applications) [4].

The implementation of PV can be considered a process which begins with the observation of vineyard attributes, followed by the interpretation and evaluation of collected data, implementation of targeted management (e.g., irrigation, fertilizers, spray, pruning or other canopy management, and even selective harvesting), and, finally, evaluation of the implemented management [5]. The efficiency of PV, particularly referring to vineyard zone delineation for site-specific phytosanitary foliar applications, depends on many interacting factors related to canopy crop characteristics like height, vegetative stage, or growing habits of the corresponding grape variety, which must be properly combined to adapt the chemical application to the foliar part of the crop [6]. In this context, georeferenced information on grapevine height at the field scale is one of the most important structural inputs used to map and monitor the vineyard and to provide accurate information for rational decision-making [7].

Phenological development stage is related to biophysical processes, and among all phenotypic characteristics, crop height is an adequate indicator of crop yield [8,9], evapotranspiration [10–12], health [13,14], and biomass [9,15]. Most of the works conducted to measure tree height or crown volume among other characteristics using geomatic techniques have been related to forest areas [16,17]. These geometric characteristics, to a lesser degree, are also used in agriculture as indicators to evaluate pruning, pest effects on crops or fruit detection [18–21], probably motivated by their difficulty to measure [22]. Collecting these data at the field scale is time-consuming and offers uncertain results because of the variability of tree crowns in orchards and the difficulty of fitting to geometric models such as cones or ovoids. To date, the measurement and characterization of plant structures has been carried out using different remote sensed alternatives like radar [23], hemispherical photography [24], digital photogrammetric techniques [25], light sensors [26], stereo images [27], ultrasonic sensors [28], and Light Detection and Ranging (LiDAR) sensors [29]. Despite the great variety of technologies used to characterize the 3D structures of plants, many of them have aspects that limit their use; only a small group of them are suitable for this purpose, with LiDAR and those based on stereoscopic images being the most relevant [30]. On the one hand, those methodologies based on terrestrial laser scanners are very precise in measuring tree architecture [31,32]; however, they are inefficient over large spatial extents [33]. Similarly, those methods based on stereoscopic images require a very high spatial resolution to properly model the 3D characteristics of woody crops [34]. In this context, images registered by sensors on board satellites or piloted aircraft platforms do not satisfy these technical requirements, with unmanned aerial vehicles (UAVs) instead being the most adequate platform. The advantages of UAV application in agriculture have been demonstrated in comparison to traditional platforms regarding their very high spatial and temporal resolution [35] and low cost [36,37], which make UAV technology an adequate tool to monitor crops at the field scale [35,38].

UAVs are capable of carrying sensors like LiDAR [39], RGB [40], thermal [41], multispectral [42], and hyperspectral [43] sensors. Although LiDAR UAV data, combined with data from Global Navigation Satellite System (GNSS) and inertial measurement unit (IMU) sensors, provide 3D point clouds to monitor plant structure information, their use is limited because of the system weight and economical cost [44], requiring very strong platforms [45]. As an alternative, UAV images registered by passive sensors can form an alternative for the 3D characterization of crops by producing digital surface models (DSMs). A DSM can be understood to be an image in which pixel values contain elevation information or a set of 3D geometries. DSMs are obtained by Structure from Motion (SFM) algorithms, which can also produce very highly dense 3D point clouds with color which corresponds to the color of the original image pixel where each point is projected as part of the processing pipeline.

In the case of agricultural applications, one of the most crucial steps is the segmentation of soil and vegetation. Using orthomosaics, the segmentation of soil and vegetation can be carried out by vegetation indices (VIs) using different spectral bands and their combinations. Of all the possible VIs, color vegetation indices (CVIs) using common red, green, and blue (RGB) sensors onboard UAV platforms are used to accentuate plant greenness [46]. On the other hand, other methods are based on using DSMs. Many approaches have been developed to detect, delineate, and segment objects in either raster or vector data [47–49]. For raster DSMs, some strategies have used object-based image analysis (OBIA) [50,51], local maxima [52,53], or watershed segmentation [54,55], among others. Of these, the OBIA methods have successfully classified and identified single olive trees [56] and vertical trellis vineyards [57], although OBIA methods need to design an effective rule set to assign the correct scale, shape, and compactness parameters to obtain meaningful objects [58]. For vector DSMs, authors have used the adaptive clustering algorithm [59] or top-to-bottom region growing approach [60], and previous research projects have successfully used DSMs on both herbaceous [61,62] and woody crops [56,57], based on geometrical approaches. Therefore, methods using vector DSMs have mainly focused on reducing the DSM to a digital elevation model (DEM), removing height value objects by a filtering algorithm. Different filter algorithms have been reported based on morphological filters [63], linear prediction [64], or spline approximation [65], with all of them based on the geometric characteristics of the point clouds and not using the color information of the points in the process.

As per the above discussion, we report in this article a new method to classify 3D UAV photogrammetric point clouds using RGB information through CVIs, tested in two vineyards on two different dates. Our specific objectives included (1) selecting the most appropriate index, (2) classifying point clouds by the selected CVI, (3) determining the heights of the vines, and, finally, (4) assessing the quality of the results obtained by comparing the UAV estimated and on-ground height measurements.

## 2. Materials and Methods

### 2.1. Study Field and UAV Flights

The presented study was carried out in two different commercial vineyards (*Vitis vinifera* L.) located in the province of Lleida, Spain (Figure 1). The first vineyard, Field A, was an area of 4925 m$^2$ cultivated with the Merlot vine variety (central coordinates 41°38′41″ N; 5°30′34″W, WGS84), while Field B was an area of 4415 m$^2$ cultivated with the Albariño vine variety (central coordinates 41°39′3.34″; 5°30′22″W, WGS84). Vines were drip-irrigated and trellis-trained and had inter-row cover crops composed of grasses. The vineyard design was focused on wine production, with the distance between rows equal to 3 m, vine spacing equal to 2 m, and north–south row orientation.

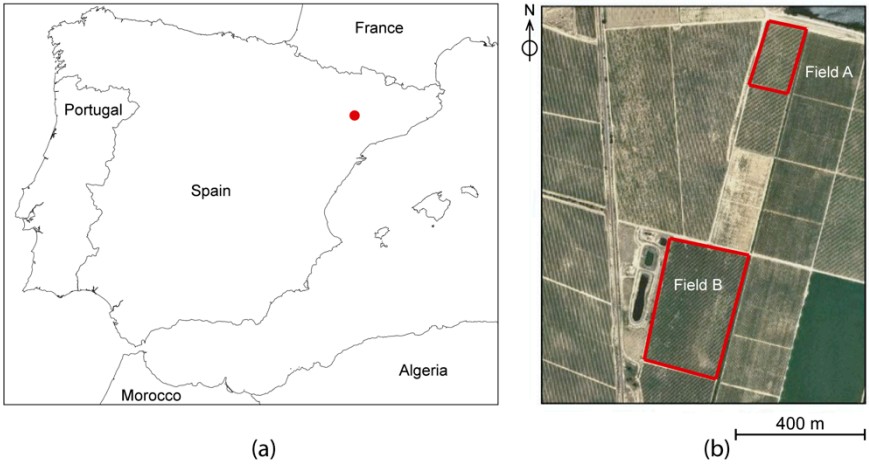

**Figure 1.** Study area: (**a**) general context and (**b**) locations of parcels.

A total of four UAV flights, two on each vineyard, was performed; the first was on 29 July 2015, and the second was on 16 September 2015, depicting two different crop stages. All UAV flights were performed under similar weather conditions. In July, the grapevine canopies were fully developed, while in September, grapes were machine-harvested. Flying at two crop stages made it possible to analyze different situations to test the validity of the proposed methodology. The UAV used was an MD4-1000 multi-rotor drone (Microdrones GmbH, Siegen, Germany). This UAV is a quadcopter with a maximum payload equal to 1.2 kg. It uses $4 \times 250$ W gearless brushless motors and reaches a cruising speed of 15 m/s. The UAV was equipped with an Olympus PEN E-PM1 (Olympus Corporation, Tokyo, Japan), which is an RGB (R: Red; G: Green; B: Blue) camera. It has a focal length equal to 14 mm. Registered images have a dimension equal to $4032 \times 3024$ pixels and a pixel pitch of 4.3 μm. UAV flights were performed at 30 m above ground level, with a ground sample distance of 1 cm and a ground image dimension of $37 \times 28$ m. Images were registered in continuous mode at one-second intervals, resulting in 93% and 60% forward and side laps, respectively. These high overlaps allow us to achieve an accurate 3D reconstruction of woody crops, according to previous investigations [40]. Five ground control points (GCPs) were placed per vineyard, one in each corner and the other in the center. Each GCP was measured with the stop-and-go technique using a Trimble GeoXH 2008 Series (Trimble, Sunnyvale, CA, USA) to georeference the DSM and orthomosaic in the photogrammetric processing.

Figure 2 summarizes the workflow for classifying points. RGB images were photogrammetrically processed to obtain a 3D RGB point cloud. From this RGB information, a CVI was calculated for each point, obtaining a 3D CVI point cloud. A sample is extracted to calculate a separation threshold value between the vegetation and non-vegetation points. Finally, this threshold value was applied to the 3D CVI point cloud to classify the points.

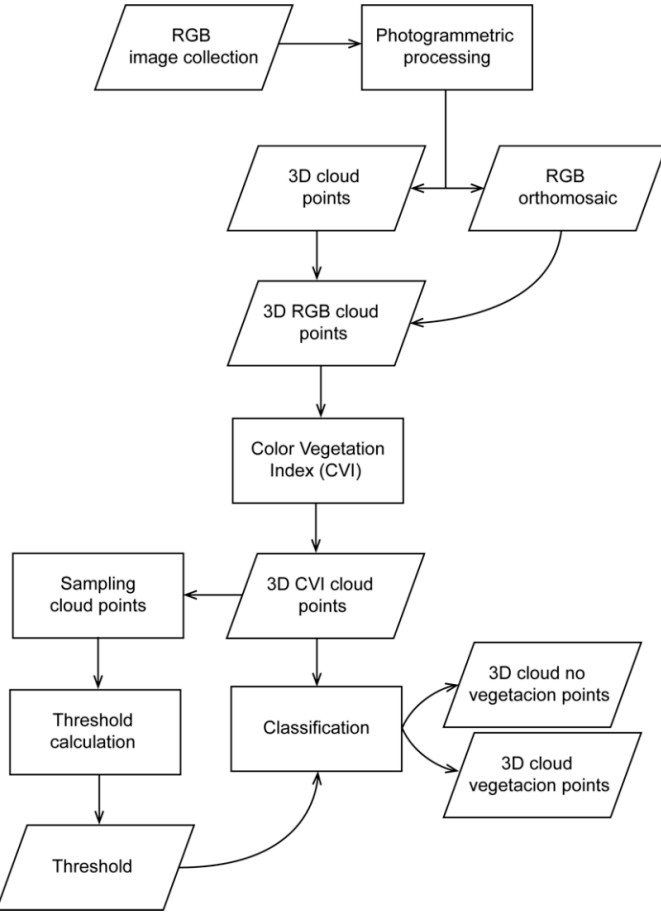

**Figure 2.** Flowchart used for the classification of points.

To produce a very dense point cloud for each UAV flight, aerial triangulation was firstly performed to determine the individual external orientation, position, and orientation of each image of the photogrammetric block. Afterwards, point clouds were generated using SfM techniques. SfM works under the principles of stereoscopic photogrammetry, using well-defined geometrical features registered in multiple images from different points of view [66]. This methodology has been validated in previous research projects [36,67], and we used Agisoft PhotoScan Professional Edition software (Agisoft LLC, St. Petersburg, Russia) for photogrammetric processing.

## 2.2. Point Cloud Classification

A crucial step of the proposed methodology is the accurate classification of those points representing vegetation and non-vegetation classes. The non-vegetation class represents the bare soil as well as the trunks and branches of vines. The classification task took advantage of the capacity of certain CVIs to enhance the discrimination of vegetation points. In this research, six CVIs were calculated based in our previous experience [46]: excess of blue (ExB), excess of green (ExG), excess of red (ExR), excess of green minus excess Red (ExGR), color index of vegetation extraction (CIVE), and Normal green–red difference index (NGRDI) (Table 1).

**Table 1.** Color vegetation indices.

| Color Index | Equation [1,2] |
| --- | --- |
| Excess of Blue | $ExB = 1.4 \cdot b - g$ |
| Excess of Green | $ExG = 2 \cdot g - r - b$ |
| Excess of Red | $ExR = 1.4 \cdot r - g$ |
| Excess of Green minus excess Red | $ExGR = ExG - ExR$ |
| Color Index of Vegetation Extraction | $CIVE = 0.4412 \cdot r - 0.811 \cdot g + 18.78745$ |
| Normal Green–Red Difference Index | $NGRDI = \frac{g-r}{g+r}$ |

[1] r, g, and b are the normalized Red, Green, and Blue spectral bands. [2] Equations are cited from [68–73].

Taking into account that each singular point has information in the RGB color space, prior to the calculation of the indices, a color space normalization for each singular point was applied following the normalization scheme described in [74]. As a result, normalized spectral components r, g, and b ranging in [0,1] were obtained according to Equation (1):

$$r = \frac{R}{R+G+B}, g = \frac{G}{R+G+B}, b = \frac{B}{R+G+B} \tag{1}$$

where R, G, and B are normalized RGB values ranging in [0,1] obtained according to Equation (2):

$$R = \frac{R}{R_{max}}, G = \frac{G}{G_{max}}, B = \frac{B}{B_{max}} \tag{2}$$

Here, $R_{max} = G_{max} = B_{max} = 255$ for 24-bit radiometric resolution.

Therefore, throughout a script developed in Matlab, the original RGB point cloud was converted to a grey-scale point cloud with the CVI value as an attribute for each of the CVIs shown in Table 1. The script has as input a LAS file of the point cloud. It reads the RGB values of each point to calculate the CVI values. As output, it generates an ASCII file for each CVI, storing the coordinates of each point as well as the index value. The potential of each CVI to discriminate vegetation and non-vegetation was evaluated by applying the M-Statistic [75] (Equation (3)), where $\mu$ and $\sigma$ are, respectively, the mean and standard deviation of both classes. Normality of distribution was evaluated by Lilliefors test. The M-statistic defines the degree of discrimination between these two classes by evaluating the separation of their histograms.

$$M = (\mu_{class1} - \mu_{class2}) / (\sigma_{class1} + \sigma_{class2}) \tag{3}$$

A value of M lower than 1 means that the histograms overlap significantly and therefore offer poor discrimination. On the other hand, a value of M higher than 1 means that the histograms are well separated, providing adequate discrimination. To calculate the mean and standard deviation for each class, a stratified systematic unaligned strategy was used as the sampling method to select points. For that, the UAV point clouds from every field and date were divided into regularly spaced regions of $10 \times 10$ m, these regions being divided into smaller sampling unit areas of $0.1 \times 0.1$ m. For each region, units with points belonging to a single class were selected manually, taking into account in-field differences. To do this, points were shown using their RGB color over top, frontal and side views. Of all the CVIs in Table 1, the one that showed the highest value in the M-Statistic test was used as the index to classify the point cloud.

Once the most appropriate CVI was selected, the next step was to determine a threshold to separate both classes. To binarize the grey-scale point cloud, a threshold value which maximizes the variance between the vegetation and non-vegetation points was chosen using Otsu's method [76]. Otsu's method analyzes the histogram of CVI values. The bimodal distribution, with two normal distributions, one representing vegetation and the remainder representing non-vegetation, was verified through the Sarle's bimodality coefficient (SBC). Otsu's methods provide an index thresholding value by maximizing the between-class variance and minimizing the within-class variance of the values. Due to the very high point density, threshold determination was performed on a sample of points from the original point cloud. This reduced point cloud was formed by reading one point out of ten (10% of the total points). All points with a CVI value equal to or below the calculated threshold were assigned to the vegetation class, while all points with a CVI value greater than this threshold were assigned to the non-vegetation class.

Matlab software (Natick, MA, USA) was employed to perform the point cloud classification and R software ((R Development Core Team, 2012) was used to perform the data analysis.

### 2.3. Validation of Vineyard Height Estimation

Vineyard height validation using point cloud classification was carried out using 40 on-ground validation randomly points distributed and georeferenced during each field and UAV flight. A photograph with the branch of the vine and a ruler was taken to measure the grapevine height at the 40 on-ground points (Figure 3). Then, the measured on-ground heights were compared to the estimated heights from the classified point clouds. For this purpose, for each position measured for every field and date, the nearest points belonging to the terrain and vegetation classes were located in the classified point cloud to later calculate the value of the height of every grapevine. The coefficient of determination ($R^2$) derived from a linear regression model and the root-mean-square error (RMSE) of this adjustment were calculated using R Commander software [77] taking into account each UAV flight independently and jointly in order to analyze the quality of the adjustment.

In addition, one-way and two-way Analysis of Variance (ANOVA) tests were applied to evaluate significant differences in height errors. The Shapiro–Wilk test and the Bartlett's test were used to assess normality and homoscedasticity of variances, respectively, to meet ANOVA assumptions.

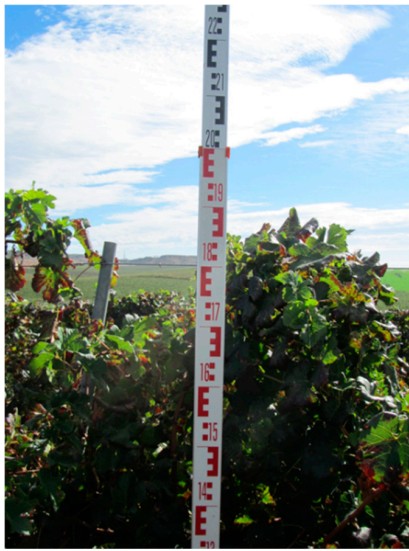

**Figure 3.** An example of vine height measurement.

## 3. Results

Figure 4 shows a partial top view of each of the point clouds generated for each UAV flight over the two fields in July and September. From visual analysis, it can be observed that the vines had a larger cross-section in July (Figure 4a,c) than in September (Figure 4b,d) because of the different phenological stages, setting (young berries growing) and harvest, respectively. In addition, the green color of the vegetation was much more intense in July than in September. On the other hand, it can be observed that, although both fields had cover crops between the crop rows, this appears to be much more widespread in Field A than in Field B. Of the four UAV flights, the one carried out on Plot B in July (Figure 4c) offers the best conditions for a manual interpretation to separate the vegetation from the bare soil, being also easier to differentiate between cover crop and vineyard.

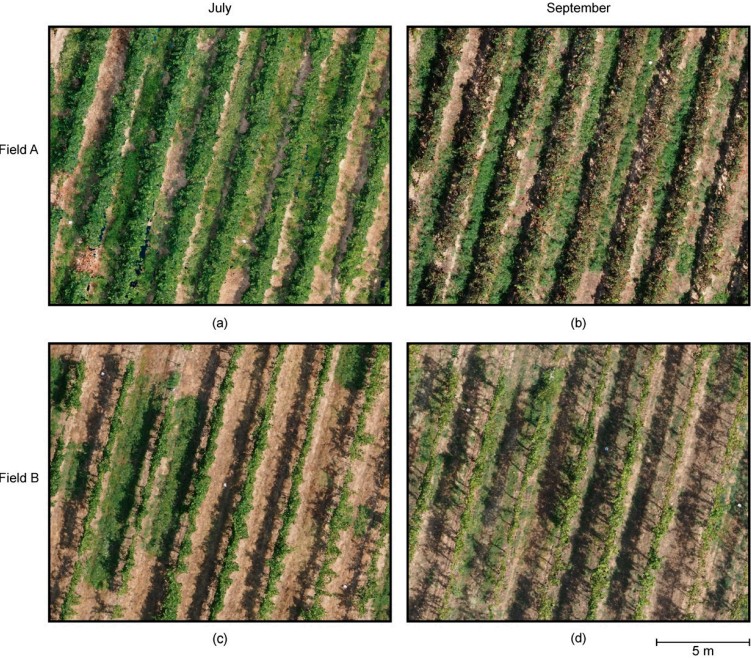

**Figure 4.** A partial top view of the point clouds generated from unmanned aerial vehicle (UAV) flights over fields A and B in the months of July and September: (**a**) Field A July, (**b**) Field A September, (**c**) Fueld B July and (**d**) Field B September.

### 3.1. Color Vegetation Index Selection

From the manually selected points belonging to the vegetation, including vines and cover crops, and bare soil classes CVIs were calculated (Table 2 shows the sample size for each field, date, and class). Firstly, the Lilliefors test was employed to examine the normality of each CVI per class, showing a p-value higher than 0.05 in all the cases. Table 3 and Figure 5 show the M-Statistic results for each field and UAV flight. All M-Statistics values obtained, irrespective of UAV flight and CVI, showed a value greater than 1. Therefore, all of them are suitable for isolating points belonging to the vegetation class from those belonging to the bare soil class. The lowest M-Statistic value was equal to 1.20, obtained in the flight over Field A in September with the ExB index (Figure 5, FA-2). On the contrary, the highest value was equal to 2.24 in Field B in July with the NGRDI index (Figure 5, FB-1). With the exception of the UAV flight over Field B in July (Figure 5, FB-1), the UAV flights showed very similar behavior in terms of the range of the M-Statistic values. In the flight over Field B in July, the M-Statistic values were the highest; therefore, the separation between classes was better. This coincides with the previous visual analysis presented (Figure 4). From the analysis of the data in Table 3, higher values of M-statistic index UAV flight over Field B was due to: a) the sum of the deviations of the two classes in the flight over Field A in July on plot A were always greater, b) same behavior appeared in September in Field B and, finally, c) the differences between the class averages in September in Field A was always smaller. This difference with respect to the September UAV flights may be due to the fact that the vegetation on this date had a less intense green color, its tonality approaching bare soil color. Therefore, the differences between the normal distributions for each class were smaller and, consequently, a lower M-Statistic value was obtained in September. On the other hand, the cover crop and vines in Field A of July (Figure 4a) presented a more accentuated green difference than did those in Field B (Figure 4b). The consequence of this is that the deviation of the CVI in the vegetation class was greater in Field A, and therefore, the value of the M index was lower. Based on the fact that the NGRDI index showed the highest values in the M-Statistic test on all flights, this index was selected to continue with for the rest of the results.

**Table 2.** Sample size of points per plot, UAV flight date and class (vegetation and non-vegetation).

| Field | Date | Vegetation [Number Points] | Non-Vegetation [Number Points] |
|-------|------|----------------------------|-------------------------------|
| A | July | 28,623 | 28,419 |
| A | September | 26,561 | 27,134 |
| B | July | 29,359 | 29,670 |
| B | September | 31,894 | 31,050 |

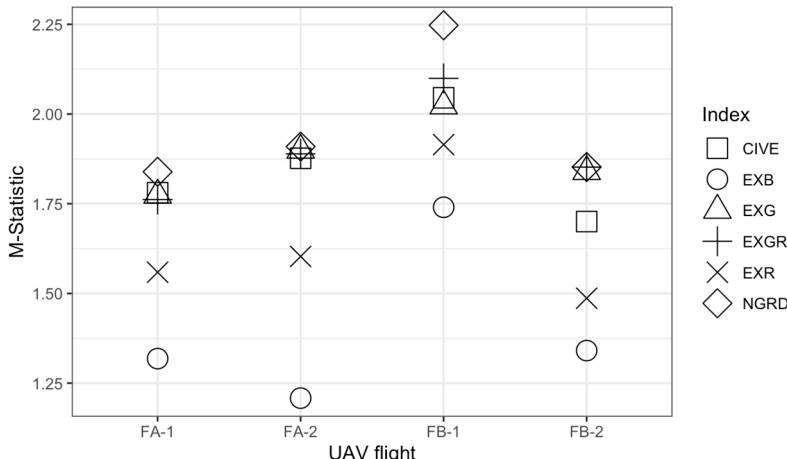

**Figure 5.** M-Statistic values for each UAV flight per Field (A and B) and month (1: July and 2: September).

**Table 3.** Mean and standard deviation (SD) of each color vegetation index (CVI) per field and UAV flight date for the vegetation and non-vegetation classes and result of the M-statistic test.

| Field—UAV Flight Date | CVI | Vegetation | | Non-Vegetation | | |
| | | Mean | SD | Mean | SD | M-Statistic |
|---|---|---|---|---|---|---|
| A-July | EXG | 0.32473 | 0.15767 | 0.0086 | 0.02055 | 1.77385 |
| | EXR | 0.04268 | 0.07627 | 0.21251 | 0.03267 | 1.55906 |
| | EXB | −0.14404 | 0.11568 | 0.04441 | 0.02727 | 1.31834 |
| | EXGR | 0.28206 | 0.22842 | −0.20391 | 0.04747 | 1.76143 |
| | CIVE | 18.66369 | 0.0635 | 18.7923 | 0.00873 | 1.78067 |
| | NGRDI | 0.11897 | 0.07445 | −0.07587 | 0.0315 | 1.83896 |
| A-September | EXG | −0.003 | 0.02099 | 0.21128 | 0.09185 | 1.89905 |
| | EXR | 0.21739 | 0.03187 | 0.09187 | 0.04643 | 1.60307 |
| | EXB | 0.05267 | 0.02171 | −0.06466 | 0.0754 | 1.20829 |
| | EXGR | −0.2204 | 0.04935 | 0.11941 | 0.13054 | 1.88897 |
| | CIVE | 18.79697 | 0.00906 | 18.70938 | 0.03758 | 1.87791 |
| | NGRDI | −0.08266 | 0.03015 | 0.06504 | 0.04723 | 1.90891 |
| B–July | EXG | 0.00393 | 0.01493 | 0.27952 | 0.12134 | 2.02242 |
| | EXR | 0.22984 | 0.02356 | 0.08768 | 0.05069 | 1.91471 |
| | EXB | 0.03238 | 0.02103 | −0.1378 | 0.10041 | 1.40132 |
| | EXGR | −0.22591 | 0.03361 | 0.19184 | 0.16538 | 2.09936 |
| | CIVE | 18.7948 | 0.00628 | 18.68292 | 0.04843 | 2.04517 |
| | NGRDI | −0.09258 | 0.02186 | 0.07311 | 0.05187 | 2.24719 |
| B–September | EXG | 0.0111 | 0.01794 | 0.21911 | 0.09513 | 1.8397 |
| | EXR | 0.20106 | 0.01717 | 0.12787 | 0.03742 | 1.34068 |
| | EXB | 0.05303 | 0.01948 | −0.10953 | 0.08987 | 1.4867 |
| | EXGR | −0.18996 | 0.03012 | 0.09123 | 0.12171 | 1.85206 |
| | CIVE | 18.78438 | 0.00726 | 18.7079 | 0.03759 | 1.7051 |
| | NGRDI | −0.06546 | 0.01514 | 0.03075 | 0.03644 | 1.86511 |

*3.2. Point Cloud Classification*

As an example, Figure 6 shows the outcome of each of the stages in the proposed point cloud classification process based on the CVI. CVI showed an SBC higher than 5/9, having a bimodal distribution with two differentiated peaks. Starting from the original point cloud (Figure 6a), the CVI was calculated to later determine the separation threshold between classes. This made it possible to have a first classification of the point cloud, differentiating between vegetation (Figure 6b) and preliminary non-vegetation (Figure 6c) points. The points classified as vegetation represented the canopy of every grapevine (Figure 6b). However, those points classified as non-vegetation (Figure 5) represented bare soil, the trunk and branches of the vine, and vegetation of both the vines and the crop cover. Those points belonging to the trunk and branches of the vines, not presenting greenish tones, together with the bare soil points, were correctly classified within the non-vegetation class. However, as can be seen in Figure 6c, there were still points that corresponded to vegetation and were classified as non-vegetation. This could be due to the fact that these points presented a color that, although green, is not as green as that of those points classified as vegetation. Thus, it was necessary to reclassify the non-vegetation class to detect new points belonging to the vegetation class. The classification procedure applied to these points began with the calculation of a new threshold based on the CVI value of these points. From this threshold it was possible to classify a new set of points belonging to vegetation (Figure 6d), differentiating them from non-vegetation points (Figure 6e). These new points classified as vegetation (Figure 6d) represented both vine and cover crop points. On the other hand, the set of points classified as non-vegetation corresponded to bare soil, as well as the trunk and branches of the vine (Figure 6e). In addition, a (reduced) number of points were still classified as non-vegetation although they corresponded to the canopy of the vines. The reason for this may be that they were

affected by shadows, presenting a different color than green. As a result, the points belonging to the vegetation class were extracted for a two-stage classification process (Figure 6f).

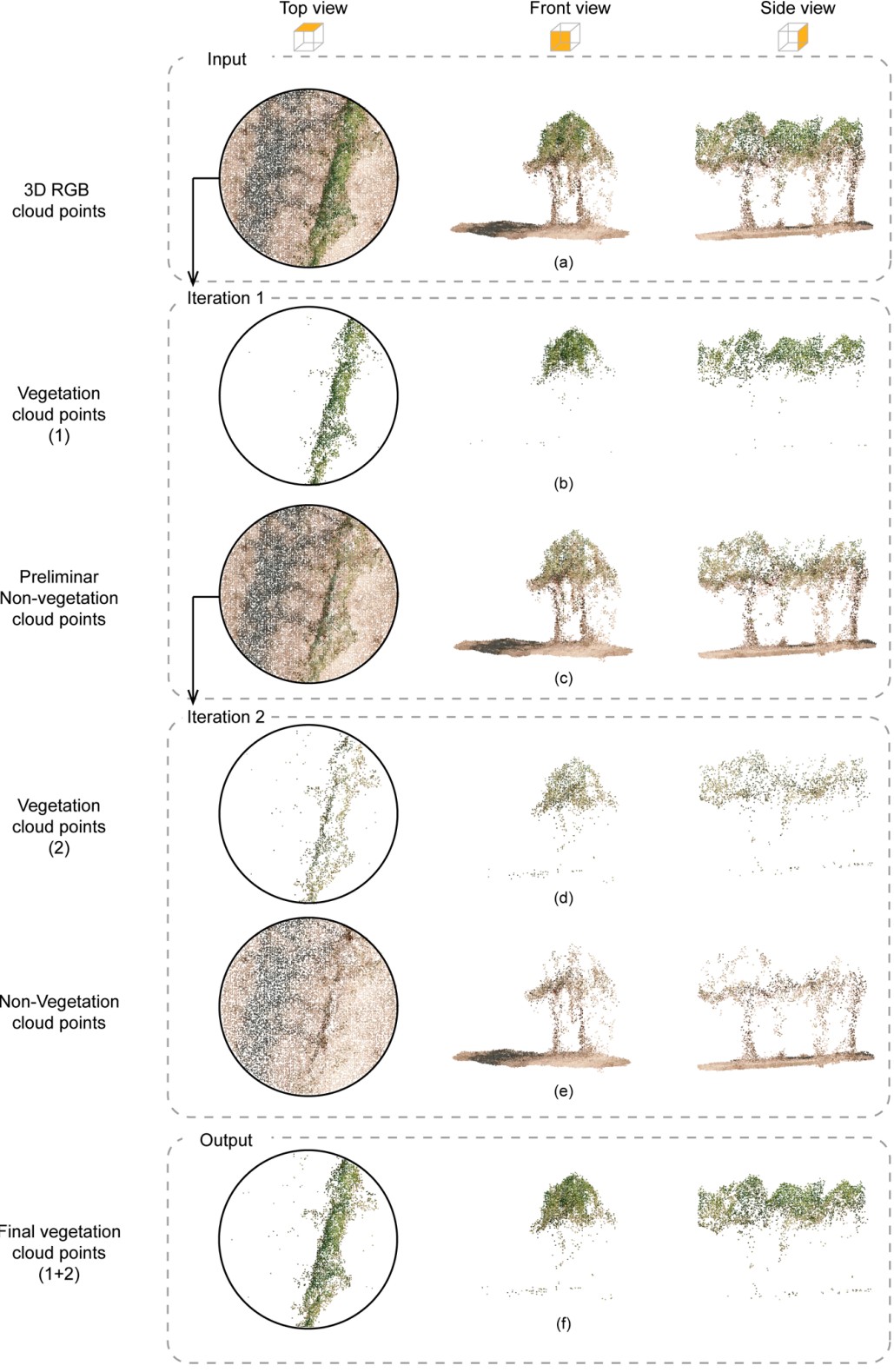

**Figure 6.** An example of the results obtained from the point cloud classification process: Cloud points: (**a**) 3D RGB, (**b**) vegetation points, (**c**) preliminar non-vegetation, (**d**) vegetation points, (**e**) non-vegetation, (**f**) final vegetation points.

### 3.3. Vine Height Quantification

Figure 7 shows the accuracy and graphical comparisons between the measured and UAV vine heights for each field and date. Independently of flight date and field, all fitted models reported a high correlation ($R^2$ higher than 0.871) and a low RMSE (lower than 0.076 m). In addition, most of the points were close to the 1:1 line, which indicated an adequate fit of the UAV estimated vine height and the measured height. The best results were obtained in the UAV flight over Field B in July (Figure 7c), matching with the visual analysis carried out on the four-point clouds (Figure 4). Shapiro–Wilk test ($W = 0.956$, p-value = 0.154) and Bartlett's test (p-value = 0.184) validated positively normality and homoscedasticity of vine height error distribution. One-way ANOVA test was used to study the differences in vine height error among the UAV flights, with the conclusion that there were no significant differences in the vine height errors among the UAV flights (d.f. = 3/156, F = 1.965, $p > 0.05$). In addition, the results of the two-way ANOVA test (Table 4) showed there was no statistically significant interaction between flight dates and fields.

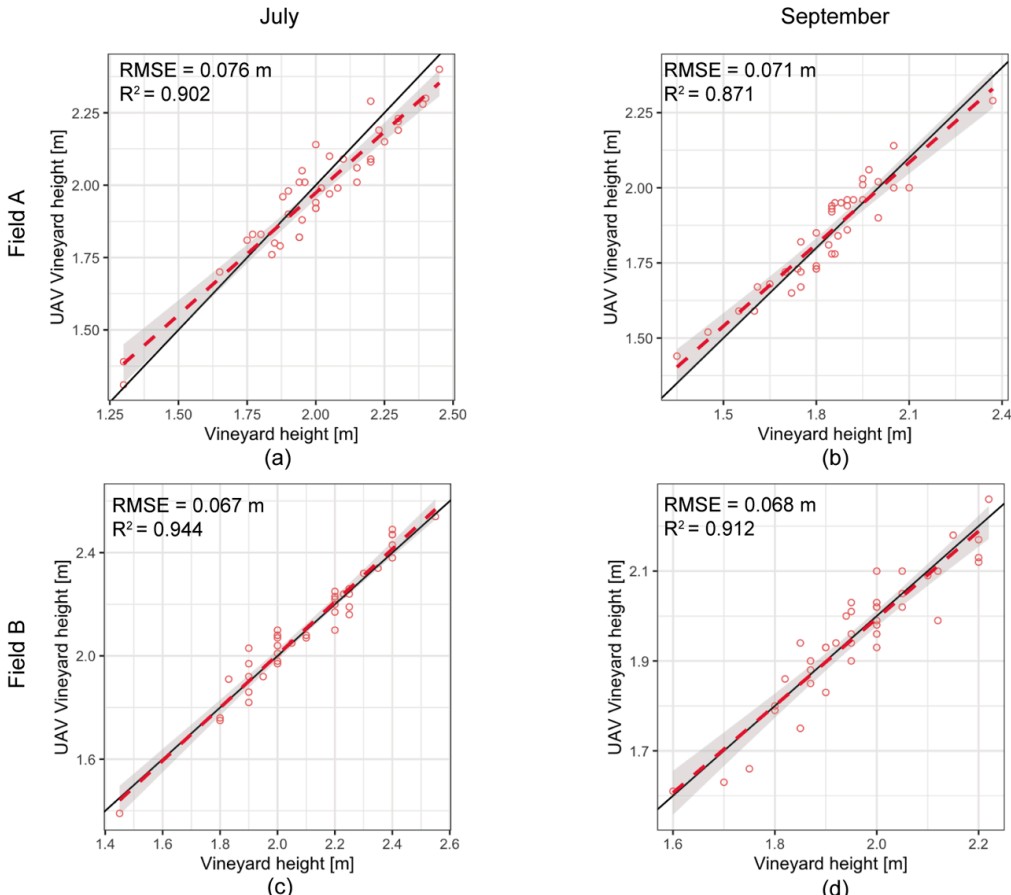

**Figure 7.** UAV vine height versus measured vine height by field and date and 95% confidence interval. The red line is the fitted linear function and the black line represents the 1:1 line. The root-mean-square error (RMSE) and the coefficient of determination ($R^2$) derived from the linear regression fit are included ($p < 0.0001$). Field A: (**a**) July and (**b**) September, Field B: (**c**) July and (**d**) September.

**Table 4.** F and *p*-values of the two-way ANOVA for vine height error.

| Factor | F | p |
|---|---|---|
| Field | 3.735 | 0.1075 |
| Flight date | 0.157 | 0.6929 |
| Field: Flight date | 2.784 | 0.1972 |

Figure 8 shows the accuracy and graphical comparisons between the measured and UAV vine heights taking into account jointly all vine heights from the four UAV flights. Linear regression model estimates a slope and an intercept equal to 0.93502 (p-value $< 2 \times 10^{-16}$) and 0.052 m (p-value = 0.006) respectively, being the residual standard error 0.063 m. The estimated plant heights of the vines showed a very high coefficient of determination ($R^2$ = 0.91) and a low RMSE (equal to 0.070 m). These results are in the same range as those obtained in other research works using other methodologies such as OBIA in the case of vineyards [57] or olive groves [40]. The points were close to the 1:1 line, indicating an adequate fit between the UAV estimated height and the on-ground measured height. Figure 9 shows a diagnosis of the linear model. Residuals were equally spread around the dotted horizontal line (Figure 9a). This indicates that there was a linear relationship between the fitted adjusted height and residuals. The normal Q–Q plot (Figure 9b) shows how the residuals were normally distributed, close to the straight line and not deviating severely. In addition, Shapiro–Wilk normality testing was applied to residuals, with a result of W = 0.974 ($p$ = 0.149). Taking into account that the *p*-value was higher than 0.05, this indicates that the residual distribution was not significantly different from a normal distribution. Figure 9c shows how the residuals were equally spread along the range of predictors and therefore have equal variance. In addition, Barlett's test, analyzing the homogeneity of variances taking into account the four UAV flights, had a p-value equal to 0.268 and therefore showed no evidence to suggest differences between UAV flights. Finally, the extreme values used were not influential in determining the linear regression model. Figure 9d shows how every residual is outside the Cook's distance and, therefore, regression results would not be altered if any measurement was excluded.

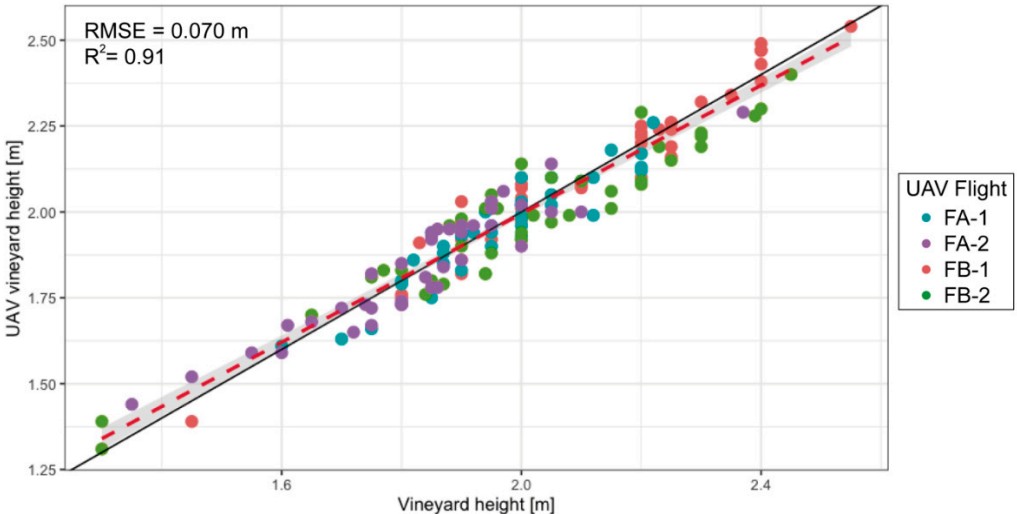

**Figure 8.** A graphical comparison of UAV estimated and on-ground measured vine heights for all UAV flights corresponding to Fields A and B and both dates (1: July and 2: September). The red line is the fitted linear function and the black line represents the 1:1 line. The RMSE and $R^2$ values ($p$ < 0.0001) obtained from adjustment are included.

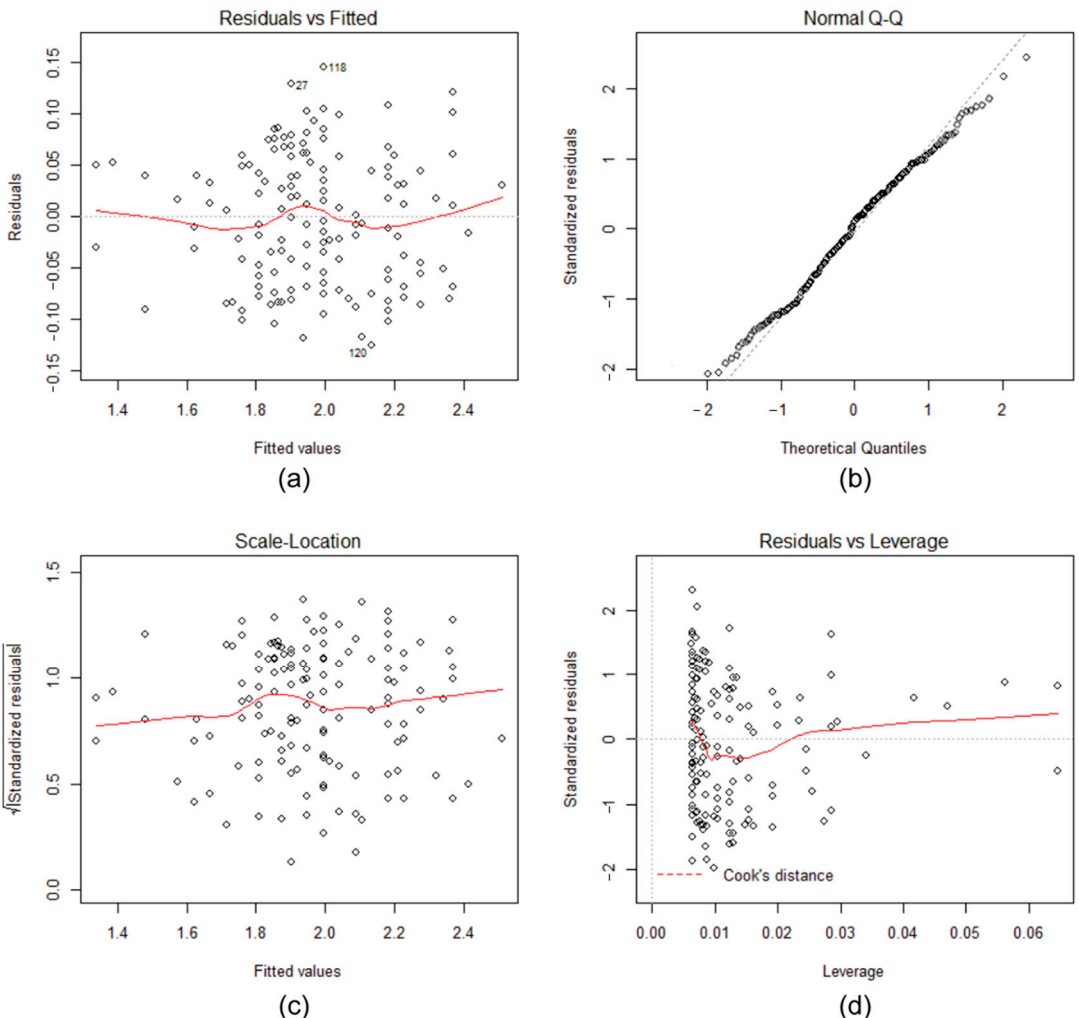

**Figure 9.** Residual analysis of the linear model of UAV estimated and on-ground measured vine heights: (**a**) residuals vs. fitted values, (**b**) normality of residuals, (**c**) scale–location, and (**d**) leverage vs. residuals.

## 4. Discussion

The results shown are slightly better than those obtained in other works using UAV flights. Other authors like [57,78,79] show $R^2$ values close to 0.8 and an RMSE equal to 0.17 m. All these authors work for the estimation of the height of the vineyard with raster vegetation height models instead of the 3D point cloud, and this may be the consequence of this slight worsening in the height estimation. De Castro et al. (2018) [57] used UAV flights for height determination based on the use of DSM and OBIA, obtaining good results supposing the rasterization of the information, with the results, therefore, depending on the size of the pixel used. On the other hand, taking into account the measurements of on-ground sensors, such as LiDAR, $R^2$ values similar to those obtained in this work are reached, but with a lower RMSE, around 3 cm [80]. Although the results obtained show a slightly higher RMSE than the on-ground LiDAR measures, it does not affect the decision making in the management of the crop [80]. In addition, using point clouds generated from UAV flights, other authors have obtained an RMSE in row height determination between 3 cm and 29 cm [81], making manual intervention necessary in the processes of classification, which could make the process less time-efficient. To solve this problem, other authors have developed unsupervised methods without manual intervention [82], which required the establishment of a series of parameters in the classification process, meaning that the quality of the process depends directly on the values selected for these parameters. Based on the

results presented herein, the use of a CVI in 3D point clouds obtained in UAV flights allows for the correct classification of vegetation and the creation of a DTM based on soil points. To our knowledge, CVIs had not previously been applied to automatically classify cloud points. The results obtained can be used later in the structural characterization of vine orchards, being necessary to validate this methodology in other woody crops. The use of CVIs as proposed in this research in the process of classifying 3D point clouds at two grapevine growth stages allowed a fully automatic method to be used, without the need for human intervention or the selection of any parameter prior to classification. Therefore, the results depend only on the radiometric information stored in each of the points without any geometric considerations like slope or distance.

Once every grapevine is classified and its height quantified (Figure 10), this information could be used for identifying smaller-sized areas for more specific in-season management (e.g., potential fungal diseases or water stress) or as a baseline to start accurate digital transformation of the vineyards for further improvement of the entire management system, avoiding laborious and inconsistent manual on-ground measurements. In addition, the proposed methodology can be used for other PV purposes. Vine volume, by multiplying the height and area covered by the point cloud classified as vegetation, or leaf area index, defined as the projected area of leaves represented by vegetation point cloud over unit of land, are potential utilities to be analyzed and evaluated in future research projects.

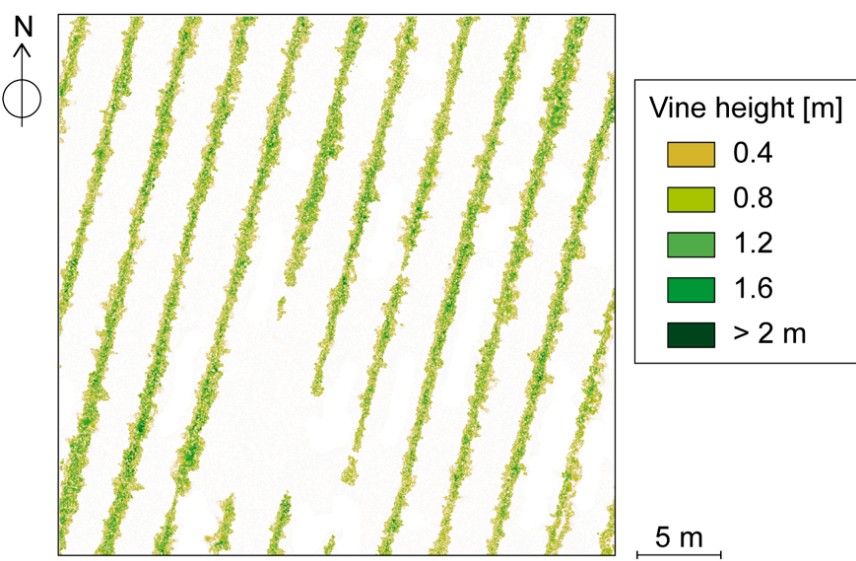

**Figure 10.** Example of a vineyard height-map based on the proposed methodology.

## 5. Conclusions

A fully automatic and accurate method was developed for the classification of 3D point clouds based only on the color information of the points; this method was then tested on two commercial vineyards at two different growth stages. Using the RGB information from each point as input for the classification algorithm, the color index value was first calculated. The threshold of separation between classes in the study area was automatically determined, allowing us to isolate those points that belonged to the vegetation class in the study area. The process was carried out completely automatically with no need to select any parameter or for previous training, eliminating any possible error inherent to manual intervention. In this way, the results are independent of the conditions and state of the crop.

The results were used to calculate the heights of the vines with satisfactory quality, as validated in two fields and at two development stages of the crop. Future work will, therefore, be dedicated to determining more structural characteristics of the vines, such as volume or crown width, to help in the monitoring of the crop and support decision-making. In addition, future work using CVIs for vines

should be used to estimate variables like biomass by constructing accurate 3D vineyard maps in each phenological phase.

**Author Contributions:** F.-J.M.-C., A.I.d.C., J.T.-S., and F.L.-G. conceived and designed the experiments; J.T.-S., F.M.J.-B. and F.L.-G. performed the experiments; F.-J.M.-C., A.G.-F., J.T.-S., P.T.-T. and F.L.-G. analyzed the data; A.I.d.C. and F.L.-G. contributed equipment and analysis tools; F.-J.M.-C. and F.L.-G. wrote the paper; A.I.d.C., J.T.-S., and F.M.J.-B. collaborated in the discussion of the results and revised the manuscript. All authors have read and approved the manuscript.

**Funding:** This research was funded by the AGL2017-82335-C4-4R project (Spanish Ministry of Science, Innovation and Universities, AEI-EU FEDER funds).

**Acknowledgments:** The authors thank RAIMAT S.A. for allowing the fieldwork and UAV flights in its vineyards.

**Conflicts of Interest:** The authors declare no conflict of interest.

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
