# Peer review of "Classification of 3D Point Clouds Using Color Vegetation Indices for Precision Viticulture and Digitizing Applications"

_remotesensing, doi:10.3390/rs12020317_

Round 1

Reviewer 1 Report

The authors present and interesting approach to determining vineyard canopy height using UAV mounted RGB cameras in combination with colour indices. But the presented methodology is not as novel as the authors present, and is lacking validation of classification results, there’s improper use of ANOVA and results could be presented better. The discussion section is missing entirely.

Comments

Line 51: The way precision viticulture is described in this first sentence does not indicate to PV being cyclical. The sentence also ends with “finally”, indicating a final step; a cyclical process does not have any final steps.

Line 58: When referring to “aerial” parts of plants, do you mean foliar? Or above ground?

Line 62: What do you mean by “plant stage”? Phenological development stages?

Line 99-100: The sentence “A DSM...” would better fit into line 91, where DSMs are first mentioned.

Line 158-159: When classes were selected by hand, did the non-vegetation class also contain cover vegetation? Since images were taken in nadir, at a distance of 30 m, how did you select branches and trees? As Figure 3 shows, those are not visible in aerial photographs. Also, since class allocations were done by hand, how did you make sure, that the vegetation class contains exclusively vine leaf area pixels?

Line 184-186: Did you also account for in-field differences? For example, Figure 1 shows a strong East-West grading in field B, especially the SE part of the vineyard has sparcer vegetation cover. Include a table detailing the means and standard deviations for both classes, vineyards and sampling dates, number of data points in each class, and m-statistic values. You have 2 varieties, 2 vineyards and 2 times. In Figure 3 there are clearly visible differences between the varieties, in Fig.3 (d) the leaves are discolored, pigment ratios have changed, be it due to age, abiotic stress (e.g. drought) or diseases (e.g. grapevine yellows). Neither of these were accounted for, but they will affect pigment ratios and leaf structure, leading to visible differences.

Line 192: Why didn’t you test for normal distributions and bimodal histograms? Furthermore, why didn’t you use the improved Otsu method (Sharma et al. 2012, ISSN: 2278-0181)? Since you selected your classes by hand, didn’t perform any data cleaning, and only assume a bimodal histogram, the imporved method could yield better, more reliable segmentation.

Line 204: You start by describing validation of cloud classification, but the paragraph deals with regression analysis, not validation. In fact, nowhere do you describe validation of the classification performed using Otsus method. Also, while the proposed method is faster, a supervised classification would yield better results. For example, a RandomForest classification scheme (using at least three colour indices and/or RGB channels) would still be comparatively fast and reliable, but would require more care to avoid overfitting.

Line 207: How were these 40 plants selected? Randomly for each vineyard? Were there 40 plants in each vineyard? Were the same plants used for both imaging sessions?

Line 212: Please add the reference for R Commander.

Line 215: Did you test for normality of distribution and homogeneity of variance? Both are prerequisites for ANOVA.

Line 218: The discussion section is missing. What is included under results is weak and should be improved. See author guidelines for more information regarding the discussion.

Line 221: In order to show differences in cross sections, Figure 3 should contain images from the same area (i.e. the same vines). Images Aa and Ab, and Bc and Bd are not of the same vines, a direct comparison is therefore not possible.

Line 244-245: Normality of distribution is only assumed, but wasn’t tested.

Figure 4: The values can be seen from the y-axis, individual data points don’t need to be of different sizes, according to their M-statistic value. The indices should be represented by different symbols (e.g. full and empty circles, crosses, triangles…) and not by colour alone, since colour-blind readers will not be able to read this figure if indices are marked only by colour. The M-statistic of all CVIs increased in Field A, but decreased in Field B (the only exception was ExB). Why?

Line 264-277: The text is misleading. Non-vegetation cloud points still contain several leaf-area pixels. Furthermore, final vegetation cloud points contain several ground and stem pixels. In Line 275 you state, that misclassified vegetation pixels are due to shadowing. Yet in Figure 3 there are clearly visible differences between varieties and sampling times. In Fig. 3(d) vine plants are markedly less green, bordering on yellow. This isn’t due to shadowing.

Furthermore, classification results are poorly described. Since classifications weren’t validated, you cannot produce a confusion matrix, and sensitivity, selectivity or any other measure of classification accuracy cannot be calculated.

Line 290: You performed a one-way ANOVA. You cannot state, that results are independent of flight date and field, because you didn’t test for that. Perform a two-way ANOVA, while taking into account normality of distribution and homogeneity of variances.

Line 292: Replace the red line with a dotted or dashed line. Also, in Fig. 6(c), were there several vines with the exact same height? The distribution on the x-axis looks like an ordinal variable, not a continuous one. Flip the x and y axes; UAV vine height is the dependent variable, the measured vine height is the reference.

Line 294: You didn’t calculate any correlations, and R2 isn’t the correlation coeficient. Also, include confidence limits in the regression plots.

Line 297: Replace correlation with coeficient of determination.

Line 300: Instead of just describing the differences between the regression lines, provide the results of a statistical comparison. Please refer to Dupont and Plummer 1998; doi:10.1016/S0197-2456(98)00037-3

Line 305: You tested the residuals for normality of distribution, but not the data. Why not? Also, the methods used in this part aren’t mentioned in the methods section.

Line 324: You cannot state, that the results of this study can be used in other orchards or woody crops, since you didnt’ test for that. You used a hierachical classification approach, but didn’t validate the results and therefore cannot state whether your models overfit the data or not.

Line 326-337: This section would better fit into the literature overview in the introduction.

Line 335-336: You still had to select classes by hand, hence human intervention. You could have used an unsupervised classification algorithm. Regardless of classification method, once the method is developed and thoroughly tested, it can be used on other data sets (i.e. vineyards), without the operator having to provide and parameters.

Line 337: Please clarify how geometric factors don’t influence your results.

Line 354-355: This statement is wrong.

Line 360: The manuscript only deals with canopy height, size and density are not mentioned or tested. This statement therefore doesn’t belong to conclusions. You could move it to the discussion and expand it.

Author Response

Thank you for the opportunity to resubmit our manuscript. Our manuscript has benefited greatly from your review and we appreciate the favorable comments from reviewer. We have revised it to clarify your suggestions. We believe our revised manuscript aligns with your comments.

The answers to your comments and suggestions are in the attached file.

Reviewer 2 Report

This paper proposed a fully automatic and accurate method for the classification of 3D point clouds based only on the color information of the points. The experimental result was reasonable. Here are some suggestions for the revision. 

Please modify the presentation of the algorithm. If there is a frame flowchart that introduces the whole technical steps, it will be more readable. Section 2.1, please introduce the process of transforming the acquired RGB image into point cloud data in detail, so that readers can have a clearer understanding. Section 2.2, please explain why six CVIs was chosen in this research, as we know that there are many combinations of color indicators in the RGB color space and many color spaces to choose from. We have seen the measured on-ground heights were compared to the estimated heights from the classified point clouds. Please add state-of-art methods compared with the proposed method in the discussion. Please keep the annotation of subfigures consistent, for example, figure 8 and other figures. The authors are encouraged to cite more up-to-date articles (Guava Detection and Pose Estimation Using a Low-Cost RGB-D Sensor in the Field; High-accuracy multi-camera reconstruction enhanced by adaptive point cloud correction algorithm; In-field citrus detection and localisation based on RGB-D image analysis).

Author Response

(The authors gave the same response as above.)

Reviewer 3 Report

Thank you for your work. Its refreshing to see that there are still researchers working using CVI in the middle of the ML, DL madness. L173-174: How did you converted in grayscale CVI point cloud? Is this a ready function of Photoscan? Give some more details please fig 3&4: The field B July case, is not in line with the rest, and this is easily identifiable from fig4, as well as the visual inspection on fig3. You will need to address that both as comments (tried to do so in L243-244) as well as in conclusions. In order for you to suggest universal application of the proposed methodology, you need to explain why there is such difference on this data set, and how this affect (minor or major) the methodology L243-244: It seems there is more sunlight. Therefore i would like to ask if the time of the day and weather conditions were similar to all acquisition days. Time of day significantly changes illumination, and forecast (clouds) the same. Fig.6: clearly, field B has better results in terms of R2, during both epochs, which should be commented. What's different between A and B fields, which affects the method ? L283: R2 of 0.871, is not considered high and seems like an outlier in your case, with all other cases being >0.9 Conclussions: please add a final height map of the vine height.

Author Response

(The authors gave the same response as above.)

Round 2

Reviewer 1 Report

While the Results section has been expanded and includes a better discussion, the section itself should be renamed to "Results and discussion".

Author Response

Thank you for the opportunity to resubmit our manuscript. We appreciate the favorable comments from reviewer. We have revised it to clarify your suggestion. We believe our revised manuscript aligns with your comments.

Comments:

While the Results section has been expanded and includes a better discussion, the section itself should be renamed to "Results and discussion".

The Assistant Editor has suggested dividing this section into two separate parts, Results and Discussion. Since October 2019 MDPI has changed the structure of the articles, being necessary to have these two independent sections.

We have introduced this change in the document, see line 359.

Reviewer 2 Report

The authors improved the quality of the manuscript significantly, I will give a "yes" to the new version. A relative reference using point cloud algorithm is suggested to add in the introduction (Real-time detection of surface deformation and strain in recycled aggregate concrete-filled steel tubular columns via four-ocular vision).

.

Author Response

Thank you for the opportunity to resubmit our manuscript. We appreciate the favorable comments from reviewer. We have revised it to clarify your suggestion. We believe our revised manuscript aligns with your comments.

Comments:

The authors improved the quality of the manuscript significantly, I will give a "yes" to the new version. A relative reference using point cloud algorithm is suggested to add in the introduction (Real-time detection of surface deformation and strain in recycled aggregate concrete-filled steel tubular columns via four-ocular vision).

Your suggestion has been taken into account. See reference 49

49        Tang, Y.; Li, L.; Wang, C.; Chen, M.; Feng, W.; Zou, X.; Huang, K. Real-time detection of surface deformation and strain in recycled aggregate concrete-filled steel tubular columns via four-ocular vision. Robotics and Computer-Integrated Manufacturing 2019, 59, 36-46, doi:https://doi.org/10.1016/j.rcim.2019.03.001.
